# Effects of Resilience and Acculturation Stress on Integration and Social Competence of Migrant Children and Adolescents in Northern Chile

**DOI:** 10.3390/ijerph18042156

**Published:** 2021-02-23

**Authors:** Alejandra Caqueo-Urízar, Alfonso Urzúa, Carolang Escobar-Soler, Jerome Flores, Patricio Mena-Chamorro, Ester Villalonga-Olives

**Affiliations:** 1Instituto de Alta Investigación, Universidad de Tarapacá, Arica 1000000, Chile; 2Escuela de Psicología, Universidad Católica del Norte, Antofagasta 1240000, Chile; alurzua@ucn.cl; 3Escuela de Psicología y Filosofía, Universidad de Tarapacá, Arica 1000000, Chile; cescobars@uta.cl (C.E.-S.); jflores@uta.cl (J.F.); pmena@uta.cl (P.M.-C.); 4Centro de Justicia Educacional CJE, Pontificia Universidad Católica de Chile, Santiago de Chile 7820436, Chile; 5Pharmaceutical Health Services Research Department, University of Maryland School of Pharmacy, Baltimore, MD 21201, USA; ester.villalonga@rx.umaryland.edu

**Keywords:** integration and social competence, migrant children and adolescents, resilience, acculturation stress, Latin-American

## Abstract

Migration in Chile has increased exponentially in recent years, with education being one of the main focuses of attention in this cultural transformation. Integration and social competence in the migrant population are determined by several factors. The aim of this study is to evaluate the potential effects of resilience and acculturation stress on the levels of integration and social competence in migrant students in Northern Chile. In total, 292 school children of both genders aged 8 to 18, from the fourth grade to senior year of high school, participated in the investigation. A subscale of the Child and Adolescent Assessment System (Sistema de evaluación de niños y adolescentes SENA) was used to assess integration and social competence. Additionally, the Child and Youth Resilience Measure (CYRM-12) and the Acculturation Stress Source Scale (FEAC) were used. The results show that integration and social competence have statistically significant and direct associations with resilience (*p* < 0.001) and indirect associations with acculturation stress (*p* = 0.009). Both constructs could be defined as protection and risk factors, respectively, and should be considered in educational contexts to favor adaptation in the integration of migrant children and adolescents.

## 1. Introduction

It is estimated that between 2010 and 2019, the displacement of migrant families reached 66% and that nearly 42.7 million people are legal and illegal residents in countries other than their country of origin. A significant percentage of this population are single-parent families led by women who, accompanied by their children, undertake long journeys looking for jobs and economic and educational opportunities that improve their quality of life [1].

Migration is a complex process that leads to different consequences for both the host country and the migrant. On the one hand, the country of origin loses human capital and on the other hand, the host country receives a cultural and socio-demographic impact [2]. Studies on migration in Latin America and the Caribbean have shown that the political, economic, and social scenarios in the region tend to make migrant families fall into poverty. Since migration is not recognized as a legitimate right by international organizations, States do not accept legislative obligations or responsibility for designing public policies that respect or protect migrants in terms of social security, employment, or access to basic services, such as health, housing, and education. Therefore, both Latin American governments and citizens consider migratory movements as a source of social and economic problems [3,4].

Chile is one of the Latin American countries with the highest growth in migratory flows since the 1990s. Its numbers doubled in 2002 with respect to the previous decade and have shown a progressive increase since then. Currently, the number of migrants is 1,492,522, representing 7.6% of the total population, with the largest percentage coming from Venezuela, Peru, Haiti, Colombia, and Bolivia [5]. Although the reasons for migrating may be political, social, economic, better opportunities, or military conflicts, the main reasons for moving to Chile are the perceptions of steady economic growth, greater security, and political stability, which have consequently positioned the country as a desirable destination compared to other Latin American countries [6]. Migrant families perceive that they could improve their quality of life, increase their social status, obtain good job opportunities, and have the possibility of sending money back to their country of origin, among others [7]. However, migrant families in Chile currently face a different scenario due to housing problems, poverty, racial discrimination, job insecurity, and difficult access to basic services (health, education, and social security) [8,9,10].

Those who are most affected by the migrant status of their families are children and adolescents. This happens for several reasons, such as family estrangement, the processes of adaptation to new cultural norms and values, different teaching systems, and learning challenges, as well as constant exposure to situations of discrimination and violence [11,12,13].

In Chile, the mean age of the migrant child and youth population is 12 years old and 41.2% of the families are single-parent families led by women. Parenting is a complex challenge because mothers must take on the economic management of the household alone. The problem is that they end up devoting more time to work than taking care of their children [14,15].

This family panorama tends to have consequences related to dysfunctional interaction patterns, and in most cases, episodes of physical and psychological abuse or neglect are observed, whose main victims are children and adolescents [9]. In addition, it has been observed that children of migrant families tend to show early signs of self-sufficiency and are capable of taking responsibility for the upbringing and education of their younger siblings, with accelerated growth and development [16]. In Chile, 54.1% of foreign families live under the poverty level and their children are often exposed to multiple risk factors or situations of vulnerability, which are particularly triggered by the discrimination and social exclusion they suffer in their schools [14]. 

Given the geographical position of northern Chile, the educational establishments of the city of Arica have maintained a constant cultural diversity of mainly Chilean, Peruvian, and Bolivian students, where cultural patterns have been unified over time. However, this diversity has increased with other groups of migrant families (Colombian and Venezuelan students), who have their own traditions and customs. Therefore, the growing cultural diversity in the education system makes the insertion of migrant students a major challenge, since it is mediated by hegemonic notions of the dominant culture and practices that, from an intersectional perspective, reproduce their segregation [17,18]. In Chile, the percentage of annual enrolment of migrant children and adolescents has increased between 2018 and 2019, representing almost 4% of the total student population. It is estimated that by 2020, there will be 300,000 migrant students enrolled in State educational establishments [19]. 

There are several studies that have shown the harmful effects of social segregation on migrants’ mental health, noting that these experiences can be especially problematic in transitional stages close to adolescence. This is because it is a critical period of development due to the processes of deconstruction or construction of one’s own identity and increased susceptibility to the evaluation and approval of others [20,21,22]. As a matter of fact, a negative evaluation of the characteristics and cultural values of migrants drive critical internal maladjustments. In these maladjustments, guilt and contempt towards themselves tend to motivate recurrent states of depression [23,24].

In relation to the above, it is estimated that the social context of migrant children and adolescents often involves risk and protective factors. Risk factors include the following: Learning new languages and customs; family separations; racism; discrimination; and xenophobia [25]. Additionally, protective factors are as follows: Sense of belonging or group affiliation, and the perception of social support and approval of their identity [21,24]. These factors can influence acculturation processes and thus positively or negatively influence mental health [26,27]. Acculturation is one of the main consequences of migratory processes, manifested through changes in the original cultural patterns after direct contact with the host cultural group [28]. According to Berry [29], there are four acculturation strategies: (a) Integration, based on the desire of individuals to maintain relationships (familial, affective, circumstantial, or superficial) with people from their culture of origin, while aspiring to create and/or maintain relationships with people from the host culture; (b) assimilation, which consists of the rejection of one’s own culture and the desire to relate to the host group; (c) separation, understood as the desire to maintain all the characteristics of one’s own culture, rejecting the host culture and its members; and (d) marginalization, characterized by ambivalent individuals who feel alienated by both cultures. With respect to the above, it has been estimated that acculturation processes can have a negative impact when the characteristics of the migrants’ relational context are hostile and involve not only experiences of racial discrimination, inequality, and social injustice, but also an emotional distancing from their families or loved ones and difficulties in integrating and participating in society [30].

The difficulties of adaptation derived from intercultural contact usually transform into a threatening or problematic experience commonly called stress due to acculturation. Acculturation stress is usually related to the hostile character of the relational contexts in the migrant population and triggers diverse mental health problems associated with depressive symptoms, anxiety, somatization, suicidal ideas, or nutritional alterations, among others [31]. The study of acculturation stress in migrants is usually focused on negative aspects mainly linked to the clinical diagnosis of psychopathologies [27]. However, when there are social realities that favor the quality of life of the migrant population and therefore their adjustment to the host culture, acculturative experiences can be transformed into a process that favors subjective well-being [32,33].

It has been found that acculturation stress in the migrant population often has effects on the quality and stability of family and social relations as a consequence of crises and changes that migration involves [34,35]. Some examples are significant disagreements and conflicts observed regarding the reorganization of family times and spaces, the distribution of parenting or work responsibilities, and the willingness to face and resolve problems or challenges, especially economic ones [36,37]. 

In relation to all of the above, migration processes can impact migrant families in various ways, with their children constantly being exposed to adverse contexts. It has been observed that the educational training stage of migrant children and adolescents is often impacted by experiences of discrimination and contextual conditions of risk, with failure and dropout being highly probable. The trend reveals that this dropout is not usually a temporary suspension of academic activities, but rather definitive, with an increasing percentage of young migrants under 16 years old playing roles as workers and providers in their homes in the Latin American region [38,39,40]. However, despite the difficulties of socio-cultural adaptation and integration and the learning challenges faced by migrant students, it has been observed that certain groups tend to successfully present curricular advances and complete their studies, recognizing a resilient attitude as an important precedent of these achievements and progress [41]. Other studies have found that the development of resilience in migrant families can be a protective factor for the mental health of their members, especially if there are children or adolescents among them, observing the cultural processes that reveal a good fit between the culture of origin and the host culture [42]. These include the importance of family ties (a stable attachment figure), significant bonds with other people considered equals (the same cultural origin or migrant status), conflict resolution skills, and a good relationship with peers and teachers. These factors are manifested, even when the context (school, family, or community) is perceived as problematic or threatening [43,44].

The socio-cultural integration of migrant children and adolescents implies a process of reconstruction of their own identity, redefining beliefs, values, roles, and personal objectives according to their new contexts. What facilitates socio-cultural integration is the existence of health, economic, and educational conditions that improve the quality of life of their families and reinforce confidence and security in the system and structure of the host country [45,46,47].

With respect to the above, the course and development of the processes of acculturation in the migrant child and youth population depend on the conditions under which and reasons why the decision to migrate was made and the way in which they are received by the host country from political, economic, social, and cultural points of view [48].

Considering that migratory flows in Latin America tend to show significant and progressive increases year after year, together with the conditions of poverty and segregation faced by migrant families, the difficulties associated with the inclusion of foreign children and adolescents, and the need for increasingly updated studies on the characteristics of the interpersonal development of the migrant child and youth population in Latin America, the aim of this study is to evaluate the potential effects of resilience and acculturation stress on levels of integration and social competence in migrant students between 9 and 18 years of age in Northern Chile.

## 2. Materials and Methods 

We used a non-experimental study with a cross-sectional predictive design, since all variables were measured at a single moment in time and the purpose of the study was to explore the functional relationship through the prediction of several criterion variables from one or more predictors [49].

### 2.1. Participants 

Two samples were used for this study: (1) One with primary school migrant students aged 9–11 and (2) another with secondary school migrant students aged 12–18, as one of the instruments has different versions, depending on the age of the participants. The participants in both samples were chosen using a non-probability sampling strategy, based on availability [50]. The primary school sample included 139 migrant students from educational establishments in the city of Arica, where 67.6% (N = 94) belong to public schools, 31.7% (N = 44) to government-subsidized schools, and 0.7% (N = 1) to private schools. The socio-demographic details are shown in Table 1. The secondary school sample included 153 migrant students from educational establishments in the city of Arica, where 78.4% (N = 120) belong to public schools, 15.7% (N = 24) to government-subsidized schools, and 5.6% (N = 9) to public schools. The socio-demographic details are shown in Table 1. The majority of educational establishments, both primary and secondary, are characterized by serving families belonging to the lower and middle socio-economic classes in the city.

### 2.2. Instruments

Demographic data were collected using an ad-hoc scale that included questions on gender, grade, and age. 

The Sistema de Evaluación de Niños y Adolescentes (SENA) (Child and Adolescent Assessment System) [51] instrument was developed by specialists in psychopathology and psychological assessment, whose purpose is to measure a wide range of emotional and behavioral problems among those aged 3 to 18. We only used the self-reported integration and social competence scale for 8–12- and 12–18-year-olds, which is composed of nine items in both versions. An item example is "I make new friends easily". The response options for each version correspond to behavioral statements on a 5-point Likert scale (1 = “never” to 5 = “always”). Higher scores indicate that the students present a set of skills that make them sufficiently competent in the field of interpersonal relationships and that they are able to perform smoothly in them. The range of each dimension goes from 1 to 5. Sánchez-Sánchez, Fernández-Pinto, Santamaría, Carrasco, and del Barrio [52] obtained adequate evidence of the validity and reliability (α > 0.7) for each of their subscales in Spain.

The Child and Youth Resilience Measure (CYRM-12) [53] is a short 12-item scale adapted from the original CYRM-28 version created by Ungar and Liebenberg [54] and designed to screen the key characteristics of resilience in children and adolescents. This scale contains items that assess the degree of resilience to adversity from the interaction between individual, relational, community, and cultural factors (e.g., “I try to finish what I start” and “My family will be by my side during difficult times”). The response options correspond to behavioral statements on a 5-point Likert scale (1 = “never” to 5 = “very much”). Higher scores suggest higher levels of resilience. Llistosella et al. [55] translated and validated the 32-item version of the Child and Youth Resilience Measure (CYRM) into Spanish, reporting adequate evidence of its validity based on the internal structure of the test and satisfactory reliability coefficients (α > 0.8). 

Sources of acculturation stress (FEAC) [56] is a 17-item scale that has been developed and validated in Chile. It was designed to ask about culturally stressful situations that have happened to students in the last six months and how much they have been affected. It is composed of three dimensions: (1) Homesickness and differences with the country of origin, with items that focus on establishing the extent to which students miss their home country and find it different from the country in which they currently reside (e.g., “I miss the friends I had in my country”); (2) school adaptation, family, and peer relations, with items that focus on the adaptations to school, family, and peers that children and adolescents face (e. g., “I have a hard time adapting to school [college]”, “My family fights more now than before [in my country]”, and "I don’t get along with the children in this country"); and (3) experiences of discrimination, with items addressing different experiences in which participants may feel discriminated against (“Sometimes I get looked at badly for being a foreigner”). The response options for each version correspond to behavioral statements on a 7-point Likert scale (0 = “I haven’t had this problem” to 6 = “it has affected me a lot”). Adequate levels of reliability have been obtained from the subscales on yearning and difference with the country of origin (α = 0.73); adaptation at school, family, and relationships with peers (α = 0.74), and experiences of discrimination (α = 0.77).

### 2.3. Procedures

The procedures were as follows:

Approval of the ethics committee of the University of Tarapacá (nº 26.2017). This study is part of a larger project of the Educational Justice Center;Forty-two educational establishments in the city of Arica were invited to participate in the study. A total of 69% agreed to participate in the study, consisting of twenty-nine establishments in total;After explaining the purpose and scope of the research, parents and then students were asked for their consent;Evaluation: The surveys were completed within 45 min. At least two trained interviewers and a teacher were present in the room. Students responded in paper and pencil format.

### 2.4. Statistical Analysis

Initially, two procedures were used to process the missing data: (1) Replacement of the missing value with the mean on the scale and (2) reporting as missing values. The first procedure was used in all instruments where the missing values were below 3%, whereas the second was only used in the acculturation stress scale (FEAC) due to the high rate of missing data. In primary school, the FEAC had a total of 36.7% missing values. In secondary school, the FEAC had a total of 27.5% missing values. Therefore, only the cases with all of the information were used for the calculation of the analyses, that is, 88 cases in primary school and 76 in secondary school.

To characterize the sample, descriptive analyses were carried out for the categorical (quantity and percentage) and quantitative (mean and standard deviation) variables. The associations between integration and social competence (SOC) and the continuous variables (age, resilience, and acculturation stress) were estimated with Pearson’s correlation. Comparisons based on the means of scores in social integration and social competence by gender were obtained through the Student’s t-test for independent samples.

Subsequently, multiple linear regression analyses were performed to examine the variables that could be associated with the social integration and social competence scores. Multiple linear regression analysis included all variables that were relevant (*p* < 0.05) in the Pearson’s correlation and Students’ t-test. The stepwise method was used to define the variables that entered the regression model. The final model, in each sample, incorporates the standardized ß coefficients, which represent a change in the standard deviation of the dependent variable. The independent variables with the largest standardized beta coefficients suggest a greater relative effect on social integration and competence. The presence of collinearity between the independent variables was ruled out by means of the inflated variance factor, which was less than 2 in all of them. All the analyses were carried out through version 25.0 of the Statistical Package for the Social Sciences (SPSS) program [57].

## 3. Results

One hundred and thirty-nine primary school students and one hundred and fifty-three secondary school students participated in this study. In primary school, the mean age was 10 years (SD = 0.8), 64 (46%) were female, and 75 (54%) were male. In terms of nationality, the majority were Bolivians (45.3%) and Peruvians (34.5%). Participants presented mean scores closer to an intermediate level for social integration and competence and acculturation stress, while they presented mean scores close to a high level for resilience. In secondary school, the mean age was 14.4 years (SD = 1.8), 79 (51.6%) were women, and 74 (48.4%) were men. In terms of nationality, the majority were Bolivians (45.8%) and Peruvians (34.6%). Participants presented means closer to an intermediate level for social integration and competence, means close to a low level for acculturation stress, and means close to a high level for resilience. The skewness and kurtosis were within the range, indicating that the variables were normally distributed [58]. Sociodemographic details are shown in Table 1, Table 2, and Table 3. 

In primary school, Pearson’s correlation analyses showed that integration and social competence have statistically significant and direct associations with resilience (*p* < 0.001) and inverse associations with acculturation stress (*p* = 0.009). Student’s t-test showed that there were no differences regarding gender for integration and social competence (boys = 3.25 [SD = 0.8]; girls = 3.46 [SD = 0.8]; *t =* 1.501; *p* = 0.136). The details of the correlation matrix for the primary school sample are shown in Table 2.

In secondary school, Pearson’s correlation analyses showed that integration and social competence have statistically significant and direct associations with resilience (*p* < 0.001) and inverse associations with acculturation stress (*p* = 0.038). Student’s t-test showed that there were no differences regarding gender for integration and social competence (boys = 3.01 [SD = 0.6]; girls = 3.22 [SD = 0.8]; *t =* 1.413; *p* = 0.160). The details of the correlation matrix for the secondary school sample are shown in Table 3.

In both primary and secondary school multivariate analyses, social integration and social competence were considered as dependent variables, while resilience and acculturation stress were included as independent variables. Age and gender were not incorporated as independent variables in the multiple linear regression models because, in previous Pearson correlation analyses, no statistically significant relationship was observed between these variables and social integration and social competence. 

In primary school, the linear regression model was statistically significant (F = 7.346; *p* = 0.001). Both resilience (ß = 0.268; *p* < 0.001) and acculturative stress (ß = -0.252; *p* < 0.005) were included the regression equation, allowing 12.7% of the variability of integration and social competence to be explained, which suggests that higher levels of resilience and lower levels of acculturative stress can lead to better levels of integration and social competence. The details of the multivariate analyses for the primary school sample are shown in Table 4. 

In secondary school, the linear regression model was statistically significant (F = 7.918; *p* = 0.001). Only resilience (ß = 0.340, *p* < 0.001) was included in the regression equation, allowing for an explanation of 11.2% of the variability of integration and social competence, which suggests that the greater the manifestation of resilient behaviors, the greater the levels of integration and social competence. The details of the multivariate analyses for the secondary school sample are shown in Table 5. 

## 4. Discussions

The aim of this study is to evaluate the potential effects of resilience and acculturation stress on levels of integration and social competence in migrant students in Northern Chile. The findings show moderate levels of integration and social competence, which suggests the existence of difficulties conditioning social inclusion and interpersonal skills development in the migrant child and adolescent population in a social and cultural context different from their origin. It is observed that resilience and acculturation stress play an important role in such integration and competence.

Several studies have found that the recognition of a different person in multicultural societies is a major challenge, due to the normalizing, hegemonic, and segregating cultural imposition of dominant societies on minorities. This does not facilitate a true socio-cultural adjustment, exacerbating the feeling of exclusion and uprooting in this population [45,46,47]. The processes of acculturation related to the capacity to confront and adapt to economic, legislative, political, and social conditions different from those of their origin represent the best way to articulate conjectures in relation to the interpersonal development of migrants [18].

In this study, we hypothesized that the geographical and historical characteristics of the context in which migrants arrive may be influencing the way in which they integrate and develop socially. The city of Arica (North of Chile) is a multicultural border zone where it is possible to observe a natural coexistence between the local and migrant population. However, the levels of acculturation stress of migrants participating in this study are likely to be due to the social gaps related to the lack of options of decent jobs, experiences of abuse or labor exploitation, economic instability, and housing precarization, along with difficulties regarding access to and the low quality of basic services, such as health, education, and social security. These conditions act as major obstacles to integration and the development of a sense of belonging or affiliation with the host culture, even triggering resistance and a lack of openness to change in the face of new norms, customs, and values, making it increasingly difficult for migrants to adapt socio-culturally [59]. 

The main reason for the social injustices experienced by migrants is the existence of a migratory institution that has not yet managed to adjust to the progressive changes following the exponential increase in the number of foreigners entering Chile each year and, therefore, has not duly safeguarded the respect and guarantee of their rights or recognized their economic, social, and cultural contribution to the country [60]. 

The problems and challenges experienced by migrant families are diverse and may increase the stress of acculturation. Some studies suggest that these difficulties may be related to Chile’s individualistic and competitive culture, moral fundamentalism, ethnocentrism, and neoliberal capitalist system, which do not encourage new visions, talents, or capabilities. These features establish exclusive criteria and selective assessments that determine the subjects or groups that may or may not form part of Chilean society, promoting the segregation of migrant populations, the increasingly significant limitation of spaces of protection or refuge, and a stagnation that does not allow social mobility [61,62], affecting the quality of life and well-being of all its members. High levels of acculturation stress are associated with feelings of isolation, rejection, identity conflicts, anxiety, psychosomatic illness, and depression, as well as with school failure or dropout. This is especially apparent when individuals are involved in circumstances or events in which they believe they have no control, will, or right. In addition, the acculturation stress may be influenced by the lack of justice and social inequalities [16,63,64,65].

The increase in migratory movements in Latin America highlights the urgency and importance of knowing the psychosocial conditions under which migrant families subsist, distinguishing what facilitates and hinders their integration into new socio-cultural contexts [66]. It has been found that economic insecurity; the denial of access to decent jobs; and the low quality health care, housing, and educational services provided by countries to their migrant populations can even be described as a human rights violation. The violation of human rights significantly hinders the development of belonging, affiliation, approval, and recognition associated with a host culture [1,67].

Another finding of interest in this study is the effect of resilience on social integration and competence in both primary and secondary education. Previous research has demonstrated the relationship between resilience and school success. Specifically, it has been observed that migrant children and adolescents who report high levels of resilience tend to present not only academic achievements similar to or better than the means of non-migrant students, but also fewer difficulties in socio-cultural adaptation and integration [40,68]. These findings are consistent with those obtained in the present study.

Resilient attitudinal dispositions in migrant students may be especially related to the quality of bonds and perceived social support in family and educational contexts. Some studies suggest that parents and teachers can be important models of resilience, favoring the social integration and participation of their children or students and the development of meaningful bonds with their peers, even when an adverse environment exists [40,67,68,69,70,71,72,73]. Although resilience is rarely the result of in-school interventions alone, the various programs/training, such as Psychosocial Structured Activities [74] ‘Be yourself and have a ball’ [75], and the Hong Kong Healthy School Award [76], converge in focusing their efforts on strengthening student engagement with school and increasing personal skills. These programs mainly promote prosocial behaviors, cultural activities, and the prevention of drug use and antisocial behavior. In addition, they provide teacher–student interactions, where the teacher is available and accessible to the students, is committed (listening to their concerns), is empathetic in difficult circumstances, and advocates for them using the support networks available to “at risk” students [77]. Other findings propose that resilience often has different characteristics, depending on the identity or cultural background of migrant students. Therefore, the development of future studies that can distinguish differential patterns and their effect on adaptation, integration, and the development of social skills in foreign children and adolescents is recommended [40,78,79,80]. 

It is important to consider the limitations of the present study. It is difficult to offer an explanatory model, whose proposal integrates other psychosocial factors that explain the variability of integration and social competence in migrants, such as the perception of social support, the sense of belonging and degree of affiliation with the host culture, satisfaction with the living conditions, perception of vulnerability, social justice, family identity, or time spent in Chile by migrant students. Therefore, it is suggested that future research pays special attention to the contextual aspects involved in the reality of the migrant population, as well as evaluates the possible mediating role that acculturation stress could be playing in the association between resilience and integration and social competence. These findings reveal the importance of developing preventive strategies from the micro to macro level that emphasize the promotion of resilient behaviors through intervention/training programs, as well as strengthen the link between schools and the needs of migrant families through internal programs led by the psychosocial team of the educational establishment. Additionally, this would make it possible to reflect on the adjustment of legislative conditions in the area of migration and the progressive social, cultural, and economic changes faced by States following migratory movements in Latin America. This emphasizes the need for new migration reforms and national and international migration policies that safeguard the freedom and rights of families migrating in pursuit of a better future in a foreign country. This will improve the quality of life of immigrants and reduce the acculturation stress that significantly interferes with their integration and social competence.

## 5. Conclusions

The difficulties faced by migrant children and adolescents in the north of Chile to achieve adjustment, adaptation, and integration with regards to this new socio-cultural context are related to acculturation stress, which can be configured as a risk factor for this population. On the other hand, and as a protective factor, resilience is configured as a positive and important factor in social integration and competence. Both acculturation stress and resilience should be addressed in terms of individual skills and school–family relationships, as well as at the level of public policies, in order to improve the social conditions of migrants’ integration in the country.

## Figures and Tables

**Table 1 ijerph-18-02156-t001:** Characteristics of the samples.

Variable		*Primary School*	*Secondary School*
		N (%)	N (%)
Gender	Girls	64 (46.0%)	79 (51.6%)
	Boys	75 (54.0%)	74 (48.4%)
Age	9	37 (26.6%)	–
	10	61 (43.9%)	–
	11	41 (29.5%)	–
	12	–	30 (19.6%)
	13	–	30 (19.6%)
	14	–	19 (12.4%)
	15	–	27 (17.6%)
	16	–	26 (17.0%)
	17	–	17 (11.1%)
	18	–	4 (2.6%)
Nationality	Peruvians	48 (34.5%)	53 (34.6%)
	Colombians	4 (2.9%)	13 (8.5%)
	Venezuelans	7 (5.0%)	6 (3.9%)
	Bolivians	63 (45.3%)	70 (45.8%)
	Ecuadorians	1 (0.7%)	4 (2.6%)
	Argentinians	3 (2.2%)	–
	Others	13 (9.4%)	7 (4.6%)
Type of school	Public	148 (53.2%)	106 (69.3%)
	Subsidized	122 (43.9%)	46 (30.1%)
	Private	8 (2.9%)	1 (0.7%)

Note: N = number of subjects and % = effective (percentage).

**Table 2 ijerph-18-02156-t002:** Means, standard deviations, and correlations between the study variables in the primary sample.

Variable	Mean	(SD)	Min–Max	2	3	4	5
1. Integration and social competence	3.3	(0.8)	1.6–5.0	−0.097	−0.127	0.306 **	−0.275 **
2. Age in years	10	(0.8)	9–11		.055	−0.187 *	0.219 *
3. Gender	–	–	–			−0.144	0.164
4. Resilience	46.3	(8.1)	18–60				−0.085
5. Acculturation stress	22.4	(16.3)	0–66				

Note: (SD) = standard deviation; Min–Max = minimum and maximum; * = *p* < 0.05; ** = *p* < 0.01.

**Table 3 ijerph-18-02156-t003:** Means, standard deviations, and correlations between the study variables in the secondary sample.

Variable	Mean	(SD)	Min–Max	2	3	4	5
1. Integration and social competence	3.2	(0.7)	1.2–4.8	0.014	−0.114	0.308 **	−0.197 *
2. Age in years	14.4	(1.8)	12–18		−0.001	−0.201 *	−0.100
3. Gender	–	–	–			−0.075	0.094
4. Resilience	45.1	(7.2)	19–59				−0.069
5. Acculturation stress	16.1	(12.9)	0–52				

Note: (*SD*) = standard deviation; Min–Max = minimum and maximum; * = *p* < 0.05; ** = *p* < 0.01.

**Table 4 ijerph-18-02156-t004:** Factors associated with integration and social competence in the primary sample.

Variable	Multivariate Analysis		
	*ß*	*ß* Standardized	*p*-Value
Intercept	2.054	–	–
Resilience	0.033	0.268 **	0.009
Acculturation Stress	−0.014	−0.252 *	0.014
R-square corrected			0.127

Note: *ß* = beta coefficient; *ß* standardized = standardized beta coefficient; * = *p* < 0.05; ** = *p* < 0.01.

**Table 5 ijerph-18-02156-t005:** Factors associated with integration and social competence in the secondary sample.

Variable	Multivariate Analysis		
	*ß*	*ß* Standardized	*p*-Value
Intercept	2.085	–	–
Resilience	0.027	0.340 **	0.000
Acculturation Stress	−0.009	−0.177	0.053
R-square corrected			0.112

Note: *ß* = beta coefficient; *ß* standardized = standardized beta coefficient; ** = *p* > 0.01.

## Data Availability

The data supporting the results of this article will be made available by the authors, without undue reservation, to any qualified researcher.

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
