# Peer review of "Effects of Resilience and Acculturation Stress on Integration and Social Competence of Migrant Children and Adolescents in Northern Chile"

_ijerph, 2021, doi:10.3390/ijerph18042156_

Round 1
Reviewer 1 Report
- This is an interesting paper that looks at a very important topic, namely at the effects of resilience and stress associated with acculturation on social competence and integration of immigrant children in Chile. I have some small comments and suggestions.
- On p. 2, l. 90 you mention “(…) often involves risk and protective factors”. I would be interested to read a bit more about what protective factors there are in this context. You mention some, but the sentence is rather long and a bit unclear
- On p. 3, l. 104-105: could you provide a reference for that?
- 3, l. 127-128: I agree that “the development of resilience in migrant families can be a protective factor for the mental health of their members”. I would ask you to be more specific: what factors lead to the resilience? We know that resilient children often show either familial protective factors (e.g., at least one stable attachment figure, constructive communication style), protective factors outside of the family (e.g., contact with peers, good relationship to teachers or neighbors) and/or protective factors within the child (e.g., ability to solve problems, characteristics that are well perceived by others, such as friendliness. Which ones do you see in the literature?
- You mention “significant bonds with other people considered as equals (…)” – this is a great specification.
- 4, Materials & Methods: can you explain what a “transversal design of correlational-causal type” is?
- 4, lines 164-167: you mention “public schools” twice, once with a sample of N=94, and then again with an N=1. Is there a typo?
- 4, l. 185/6: you mention “a higher score indicates a greater presence than the scale measures” – what do you mean by that?
- General question: how long have the participants been in Chile? Might that duration be a moderator/mediator?
- “Resilience”, “Integration” and “Acculturative Stress” do not need to be written with capital letters, I think.
- In your discussion, could you suggest how the resilience of children could be strengthened? Are there any trainings / programs that exist already? Is there a type of behavior that teachers could display that would support children?
- Also in your conclusion: how could the social conditions be improved?
- General comment: Many sentences are very long. Shortening them would make it easier to understand the sometimes complex matters that you are explaining.
Reviewer 2 Report
Overall, this is a well written paper. The main area for improvement is the Discussion section where I feel the results of the study and implications of the findings are not highlighted as much as it could be. My specific comments below:
- Line 218, Please write the ethics approval number or reference number.
- Line 221: Should the sentence read, ‘consisting of’ twenty-nine establishments rather than ‘considering’
- Line 242: t student test should be written as ‘Student’s t-test’
- Lines 273 & 274 please write the p-value as: <0.001 (or appropriate value) instead of 0.000
- Can the authors discuss the main reason for the high rate of missing data for the acculturative stress scale and the implication this may have on the findings? Are there differences among the participants that had data compared to those that did not have data?
Line 283: Please revise the sentence so that it is clear which participant group is being referred to: “In secondary school students, bivariate analyses showed that, as in primary school students.”
- Table 1 shows that acculturation stress was higher among primary school students vs secondary school students – can the authors discuss the potential reasons for this?
- Discussion section should highlight the findings among the child and adolescent age groups and discuss previous research among children and adolescents more
- Results showed that among primary school students, a higher proportion of the variability of integration and social competence was explained by resilience compared to secondary school students (12.7% vs 10.4%) - can this be discussed a bit more?
- Significance of the study needs to be highlighted more and implications of this research also needs to be discussed in more detail
Author Response
Please see the attachment. Tahnk you.

Reviewer 3 Report
Thanks for allowing me to review this paper.
The paper focuses on an important issue of integration of migrant families in Chile, but the same applies to many other countries affected by migratory waves.
While the design is sound, the manuscript must be improved for publication. The data analysis is rather non-standard or explained in such a way that is very confusing (see details below) and the overall argument fails to provide a detailed context for those not familiar with the area and national socio-political context.
Overall I believe the paper is worthwhile publishing, but it needs another round of edits. Please note some grammatical/spelling errors, therefore it will benefit from a full copy-editing. A more detailed account of the different sections is provided below.
Introduction:
The introduction has some indication of the political context, but it does not explain why people move in Chile from neighbouring countries, especially when the authors suggest that there are no opportunities for migrants? My hunch is that the profile of migrant families is very important here and sets up the scene for the individuals to migrate, even when opportunities are limited. To make a comparison in the middle east, Syrian migrants running away from their own countries because of the war and regime, or the Rohinga muslims in south east asia who are persecuted. I’d like to know more of what are the drivers for migration to Chile.
Then there is a jump to ‘cultural diversity’ without explaining what this refers to: as a reader with limited knowledge of the context I’d guess that Spanish speaking migrants will have a better chance than Caribbean migrants? Again, this is a generalisation resulting from the lack of clarity about the context described.
The psychological factors describing the migrants families is ok, but again, it would be better anchored in the context and previous literature with examples rather than too generic statements. The issue of identity, risk factors, and pragmatic issues of day-today needs (food, shelter, jobs etc) are well known, but are the target participants into a category which can be classed under the poverty line, or are the families in a ‘middle class’ and therefore we could say that basic needs are secured first?
Methodology
The description of the tools used would also benefit form more detail (I’d suggest to include the questions in the additional material rather than the consent forms) and provide a more thorough account of the scales (with reference to the theory presented in the introduction) and why they were selected.
Analysis
The start of the analysis is clear, up to the regression models, in which it is not clear what is referred to as ‘bivariate analysis’ especially what measures have been used and how.
If I’m trying to match the clear description in section 2.4 and the results in section 3, this is where I’m getting lost quickly.
From the description, as a first step I was expecting a summary correlation table between scales in the primary and secondary samples, then a clear indication of what are the variables predicted by the regression model and which variables are used as grouping factors (it seems that gender is the only brought in to look for differences, but why? What about other factors as age, nationality and type of school?) is the type of school implicitly underlying a socio-economic status?
There are weird notations in the presentation too: when referring to teliability of the scales, Cronbach alphas are expected (in the instruments CYRM is shown with w >0.8 with no explanation why)
Conclusion/discussion
In the discussion, the authors answer some of the questions I raised in the comments about the introduction, particularly the broadening of the socio-political context and tapping into the philosophical aspect, focusing on motivation, identity and values.
I’d like to see more detail in the discussion about the psychological dimensions, and what these means in relation to the system, education and policies, but with a clear description of the impact on individuals and their families.
If the stereotype of the ‘outsider’ at school holds ground, and the individual feel rejected, what psychological measures are taken to tackle bullying, antisocial behaviours and family support?
Round 2
Reviewer 3 Report
The authors provided extended responses to all comments from the reviewers and modified the manuscript accordingly raising both the quality and clarity of the presentation.
While the authors provided better explanations of the context, i still feel that as an outsider to South America, the true reality of migration and cultural differences are not rendered enough.
Used to a European multi-cultural environment in which language is not the only barrier, but centuries of domination by colonial/imperial countries dominated the geopolitical landscape, unfortunately i fail to fully understand the differences in latin-american countries which have indigenous populations, all speak spanish and have received waves of migrations from Europe and Africa since the 1800s. This makes it hard for a reader without a full knowledge to appreciate what is intended as the 'establishment' and why migrants would find it hard to place themselves.
part of me think that removing the political a agenda and focusing on a very clear definition of acculturation may make a better/stronger argument, but i think that more detail will be worthwhile.
Results:
i'm still confused about the reporting of some of the data: 'bivariate analysis' implies tests between pairs of variables, but this does not make sense at the outset as the design is multivariate by definition (i.e. the study explores relations between multiple variables at the same time, so multiple t-test or multiple correlations would not be the way of controlling for the partial correlations of interaction effects). if these are presented as post-hoc tests then it would be ok, but then appropriate methods to correct p values would be required (i.e. Bonferroni or Turkey for multiple t-tests, etc)
my hunch is that the authors refer to the SPSS command 'bivariate correlations' which basically produced a correlation tables between all variables included. the resulting correlation matrix does not really account for interactions and i would definitely report these as a correlation matrix rather than mixing it with the regression.
in the multiple linear regression please make it clear what is predicted and what are the predictors included; while i think that 'integration and social competence' is the metric to be predicted by resilience and acculturation stress, the decision seems to be totally arbitrary; is a model in which there are mediating factors possible?
i.e. resilience -> acculturation stress -> integration
as a suggestion, a hierarchical regression model would enable to extract the level of importance via the magnitude of the predictions, or even better using the mediation macro for SPSS (https://www.youtube.com/watch?v=y8DybnUoWoo) will be useful to aid the interpretation and conclusions
discussion and conclusion:
this is much improved, but i think that the actual explanations and interpretations follow more from the detailed socio-cultural premises rather than the psychological factor in the results and from the pragmatic point of view, shaping intervention programs need to rely on the prediction (or mediation model). to be blunt, if there is statistical evidence that by improving programs focusing on resilience has a 3x magnitude effect in reducing stress and improve integration, that's a very clear message for policymakers and schools.
while it is the desire of every author to address reviewers' comments and get published, i sincerely hope that these comments will help strengthen the quality of the analysis and the effectiveness of the argument presented
